# Mechanical and Electrical Properties and Electromagnetic-Wave-Shielding Effectiveness of Graphene-Nanoplatelet-Reinforced Acrylonitrile Butadiene Styrene Nanocomposites

**R. B. Jagadeesh Chandra** [1] , **B. Shivamurthy** [2,*], **M. Sathish Kumar** [1], **Niranjan N. Prabhu** [2] and **Devansh Sharma** [3]

1 Department of Electronics & Communication Engineering, Manipal Institute of Technology, Manipal Academy of Higher Education, Manipal 576104, India
2 Department of Mechanical & Industrial Engineering, Manipal Institute of Technology, Manipal Academy of Higher Education, Manipal 576104, India
3 Department of Materials Engineering, Indian Institute of Science, Bangalore 560012, India
* Correspondence: shiva.b@manipal.edu

**Abstract:** Polymer nanocomposites have attracted global attention as a metal replacement for electrical and electronic applications. Graphene nanoplatelets (GNPs) are widely used as a nanoreinforcement to enhance the functional and structural properties of thermoset and thermoplastic polymers. In the present study, ABS nanocomposites were prepared by reinforcing 3–15 wt.% GNPs in steps of 3 wt.%. The neat ABS and ABS+GNP nanocomposite specimens for the mechanical test were prepared using injection molding, followed by extrusion, as per American Society for Testing and Materials (ASTM) standards. It was found that the modulus of ABS improved due to the reinforcement of GNPs. Additionally, we noticed higher thermal stability of nanocomposites due to the faster heat-conducting path developed in the nanocomposites by the presence of GNPs. However, observed agglomeration of GNPs at higher concentrations and poor wetting with ABS led to the deterioration of the mechanical properties of the nanocomposites. Moreover, 350 μm thick nanocomposite films were manufactured by compression molding, followed by the extrusion method, and we investigated their electrical conductivity, magnetic permeability, permittivity, and electromagnetic-wave-shielding effectiveness. The developed nanocomposites showed improved conductivity and effective electromagnetic wave shielding by absorption. The 15 wt.% GNP-reinforced ABS composite film showed a maximum shielding effectiveness of 30 dB in the X-band.

**Keywords:** nanocomposites; acrylonitrile butadiene styrene; graphene nanoplatelets; tensile properties; electrical conductivity; permeability; permittivity; electromagnetic-shielding effectiveness

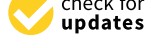



## 1. Introduction

The increased use of electronic devices/instruments/equipment in information and communication systems, the internet of things (IoT), and smart-system operations increased electromagnetic radiation (EMR) emissions. This led to electromagnetic interference (EMI) pollution in the environment. The mutual EMR interference is caused due to nearby electronic devices, instruments, and equipment creating malfunctioning of normal performance or even breakdown of other electronic devices, instruments, and equipment in the vicinity [1]. The EMI is disruptive and jeopardizes electromagnetic communication and human health. Hence, EM shielding is necessary for electronic devices/equipment to reduce EMI pollution. If the shielding effectiveness by the material is below 10 dB, it is considered little, or no shielding effectiveness and 10 to 20 dB is considered the minimum effective range. If the shielding is more than 20 dB, it is considered effective shielding and suitable for commercial applications.

Acrylonitrile butadiene styrene (ABS) is widely used in electrical, electronics, and automotive industries due to its multifunctional properties, chemical resistance, light weight, and ease of fabrication. It is demonstrated in the literature that the addition of a judicious quantity of suitable nanofillers to ABS alters the structural and functional properties of the resultant ABS nanocomposite [2]. In this direction, researchers developed lightweight, electrically conductive polymer nanocomposites consisting of an engineering-polymer matrix reinforced with carbon-based conductive nanofillers. Carbonaceous materials, such as graphene [3–5], graphene nanoplatelets (GNPs) [6], reduced graphene oxide (rGO) [7], multi-wall-carbon nanotubes (MWCNT) [8], graphene nanoflakes (GFs) [9], and carbon black (CB) [10], were used as fillers in the development of EMI-shielding polymer nanocomposite materials for electronic equipment, wireless communication devices, and aerospace applications [11–13]. They also demonstrated improved thermal stability of such composites compared to neat polymers.

Sachdev et al. [6] developed composites with ABS/graphite pellets and found 15 wt.% graphite-filled ABS has a maximum shielding efficiency of ~60 dB of 3 mm thick sample in the X-band frequency and DC conductivity of 0.166 S/cm. However, observed ABS/graphite composites are more brittle. Foam composites are attractive in EM wave shielding due to their structure and light weight. Lei et al. [1] developed ABS/graphite foam (GF) and ABS/compressed graphene foam (cGF) composites and found ABS/50% cGF composites showed 42.4 dB shielding effectiveness with a density of 0.14 g cm$^{-3}$. Vidakis et al. [2] compared ABS/GNP and ABS/CNT nanocomposites developed by filaments with various concentrations of nanofillers (0.5 wt.%, 2.5 wt.%, 5 wt.%, and 10 wt.%). They reported, due to the addition of CNT, an almost 60% increase in the tensile strength and modulus compared to pure ABS, whereas a 20% reduction in the tensile strength and modulus was noticed due to the addition of 2.5 wt.% GNPs into ABS. This was due to the physical structure of the filler shape and size. Dul et al. [14] developed ABS/GNP and ABS/CNT nanocomposites by loading GNPs up to 30 wt.% and CNTs up to 8 wt.%. They reported slightly higher stiffness and creep stability of ABS due to the addition of GNPs compared to CNT. They observed that due to the addition of 2–8 wt.% of GNPs in steps of 2 wt.%, the tensile strength increased from 39.9–41.4 MPa. The tensile strength improved from 34.1–45.1 due to the addition of CNTs. The addition of 2 wt.% CNT in ABS nanocomposites showed much lower electrical resistivity than 8–12 wt.% ABS/GNP.

Recently, Cao et al. [15] proposed a green EMI-shielding concept in which the incident microwaves are strongly absorbed rather than reflected and transmitted. Further, Wang et al. [16] were inspired by nature and developed a $Co_3O_4$@WSe2-MWCNTs nano–micro "vine" with a hierarchical structure and found efficient green EMI shielding. Additionally, the vine structure performs as a supercapacitor. They constructed a multifunctional microwave conversion and storage device. It converts EM radiation into stored useful electrical energy.

From the above literature, it is noticed that developing an effective EMI-shielding material with reasonable mechanical properties is a challenging task. In addition, extrusion, compression molding, and injection molding are the cost-effective and widely used standard manufacturing methods for mass production in plastic industries. However, very few works on EMI-shielding polymer nanocomposites were reported based on these methods. Additionally, as per the authors' knowledge, ABS+GNP nanocomposites prepared by extrusion, followed by the compression-molding process, have not been attempted to investigate their structural and electronic properties, including for EMI-shielding applications. Hence, the authors attempted to develop ABS+GNP nanocomposites by extrusion, followed by compression and the injection-molding process, followed by investigating their mechanical and electrical properties and EMI-shielding effectiveness.

## 2. Materials and Methods

### 2.1. Material

S-grade GNPs (surface area: 80–200 m$^2$g$^{-1}$, average lateral dimension: 1–5 μm, thickness: 5–10 nm, bulk density: 0.166 g cm$^{-3}$) were used as filler material. They were produced by the chemical exfoliation method with a purity of 99% and supplied by Adnano Technologies Pvt. Ltd., Shimoga, Karnataka, India. The general-purpose SD-0150-grade ABS was supplied by Samsung India Pvt. Ltd., Mumbai, and used as matrix material. The key properties of ABS are depicted in Table 1.

**Table 1.** Key properties of SD-0150-grade ABS [17].

| Property | Test Method | Test Condition | Unit | Value |
|---|---|---|---|---|
| MFI | ASTM D1238 | (200 °C/5 kg) | Gr/10 min | 1.7 |
| Izod impact strength | ISO 180 | 1/8 inch | Kj m$^{-2}$ | 25 |
| Tensile strength at yield | ASTM D 638 | 5 mm/min | Kgf cm$^{-2}$ | 440 |
| Elongation at break | ASTM D 638 | 5 mm/min | % | 40 |
| Flexural strength at yield | ASTM D 790 | 2.8 mm/min | Kgf cm$^{-2}$ | 640 |
| Flexural modulus | ASTM D 790 | 2.8 mm/min | Kgf cm$^{-2}$ | 20,500 |
| Vicat softening temperature | ASTM D 1525 | 5 kg | °C | 99 |
| Rockwell hardness | ASTM D 785 | | R-scale | 104 |

### 2.2. Fabrication of ABS+GNP Nanocomposites

Initially, GNP-reinforced ABS granules are made using the melt compounding method. Further, the nanocomposite bulk samples are manufactured using ABS+GNP granules through injection molding. The nanocomposite films were fabricated by the compression molding method using ABS+GNP granules. The detailed process is given below.

A 2.5 kg batch size of compounded granules was prepared by mixing and extruding ABS granules with 3, 6, 9, 12, and 15 wt.% GNPs in a lab-scale co-rotating intermesh twin-screw extruder (L/D: 40, D: 30 mm). The temperatures of the extrusion zones were around 190–220 °C, the die temperature was around 225 °C, and the speed of the extruder screw was maintained at 180 rpm. Further, neat ABS and the GNP-reinforced ABS granules were hot pressed in a 10-ton capacity hydraulic press at 3 bar pressure with a temperature at about 220 °C to produce approximately (200 ± 10) μm thickness films.

As per American Society for Testing and Materials (ASTM ) standards, the specimens required to investigate mechanical properties were prepared using ABS and the GNP-reinforced ABS granules in an 80-tonne-capacity injection-molding machine. Figure 1 shows the production of ABS+GNP nanocomposite samples and films.

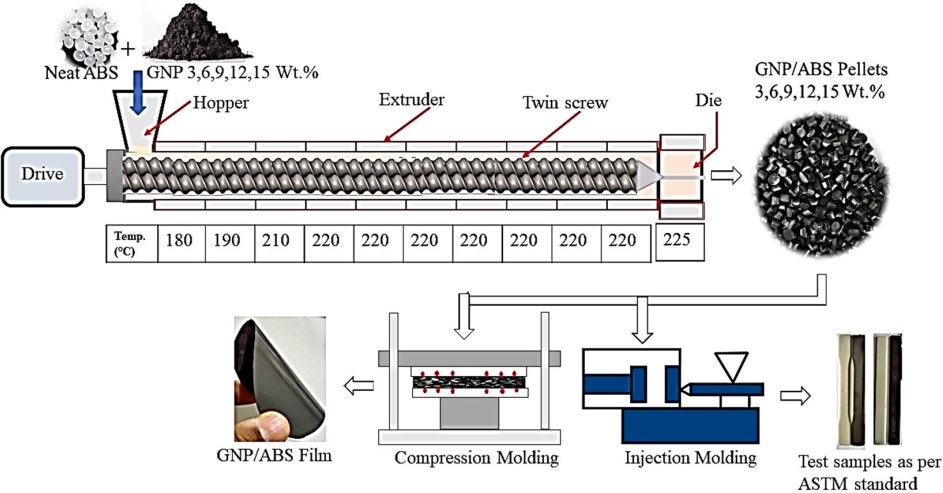

**Figure 1.** Schematic diagram of ABS+GNP nanocomposite film and test sample preparation.

### 2.3. Characterization

A scanning electron microscope (Hitachi SU 3500) was used to investigate the particle size and shape of the GNPs. The structural features of GNP and ABS+GNP nanocomposite film samples were examined by a Rigaku MiniFlex X-ray diffractometer using Cu Kα radiation (λ = 1.54 Å) with tube voltage and current at 40 kV and 30 mA, respectively.

The tensile and flexural properties of neat ABS and ABS+GNP nanocomposite samples were studied as per ASTM D 638 [18] and ASTM D 790 [19], respectively. The impact strength was investigated as per ASTM D 256 [20]. The thermal properties of all the composite samples were estimated using thermogravimetric analysis. Real and imaginary parts of complex permittivity (ε) and permeability (μ), and EMI shielding by various mechanisms of all the nanocomposite films, were estimated at the X-band (8–12 GHz). The scattering parameter (S-parameter) describes the relationship between the input–output power measured in decibels at the X-band frequency range. These were measured using a network analyzer to ascertain the EM wave reflection and transmission ability of the developed nanocomposites.

### 3. Results and Discussions

#### 3.1. Morphology of GNPs

The morphology of gold sputter-coated GNPs was investigated using a Hitachi SU 3500 scanning electron microscope (SEM) at accelerating voltages of 10.0 kV with a magnification of 8.0k. Figure 2 shows the SEM image of the GNPs. It is found that the lateral dimensions and plate-type structure are in agreement with the supplier's data.

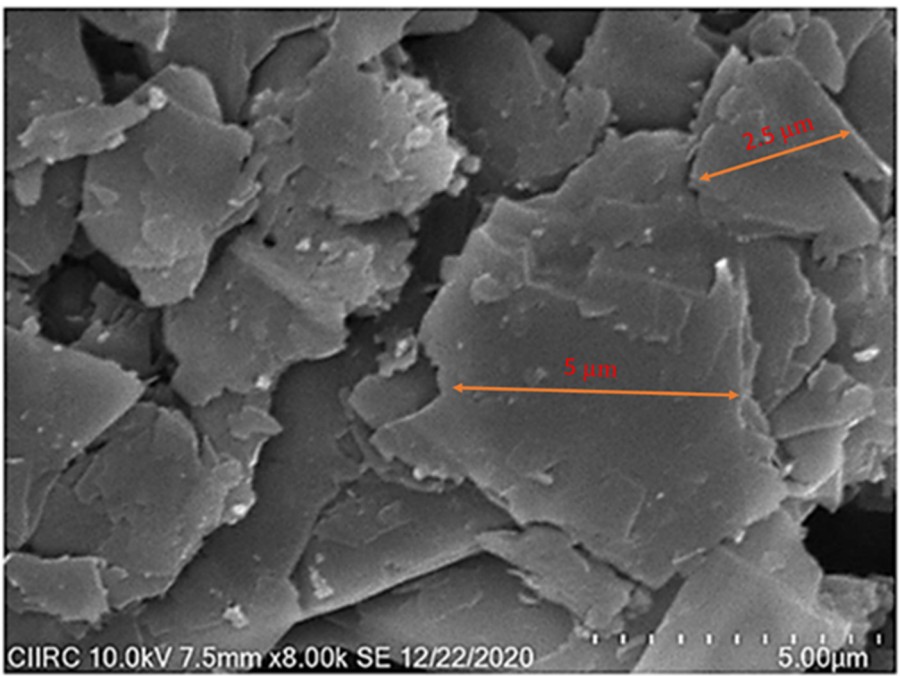

**Figure 2.** SEM image of GNPs.

#### 3.2. X-ray Diffraction

The fundamental evidence of the quality and the presence of GNPs in ABS nanocomposites was investigated by wide-angle X-ray diffraction (XRD) patterns of GNP powder, neat ABS, and ABS nanocomposites films on a Rigaku MiniFlex 600 powder diffractometer with Cu Kα radiation at a scanning rate of 1° per minute.

The XRD spectra were recorded in the 2θ range of 0–60° and are presented in Figure 3. It is confirmed from the spectra-intensive peaks at 2θ = 26.4° and 54.5°, corresponding to

the planes (002) and (004), respectively, that the materials contain graphene. The crystallite size (T) of GNPs was estimated by using the Debye–Scherrer equation [21]:

$$T = \frac{k\lambda}{\beta cos\theta}$$

(1)

where β is the FWHM, λ is the wavelength, θ is the diffraction angle, and K is the shape constant. The relative size of the GNPs was 38.75 and 33.48, obtained from the calculations using β = 0.21, θ = 13.2 °C, and λ = 0.15406 nm and β = 0.26, θ = 27.5 °C, and λ = 0.15406 nm, respectively. The interlayer spacing between the graphene d-spacing was found to be 3.374 Å.

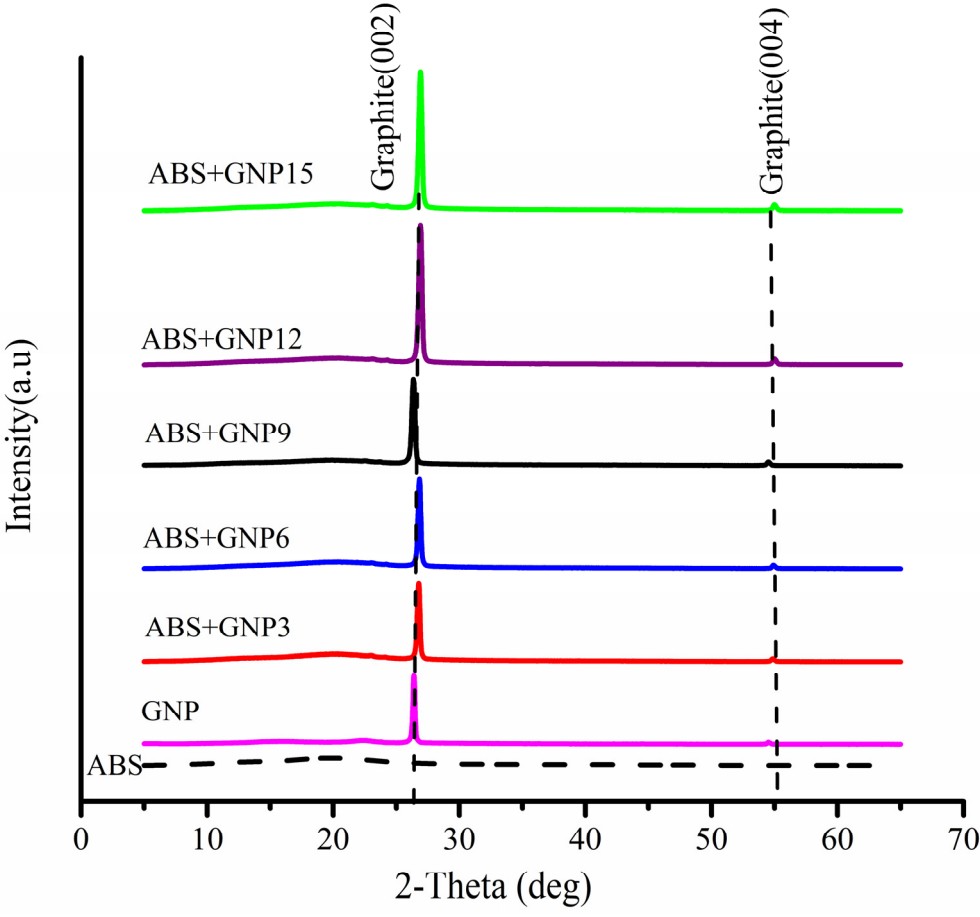

**Figure 3.** XRD spectrums of GNP, neat ABS, and ABS+GNP nanofilms.

The XRD pattern for neat ABS, which comprises a broad peak at 19.9°, is one of the characteristics of the amorphous structure of ABS [22]. No other peaks reflected in the neat ABS spectra indicate that the ABS used in this study is free from additives. The spectra of ABS+GNP nanocomposite samples showed that the peaks at 26.4° and 54.5° confirm the presence of graphene in the ABS+GNP nanocomposite film [23]. In addition, the concentration of GNPs is observed to have a narrow peak at 26.4°.

### 3.3. Mechanical Properties

3.3.1. Tensile Behavior

The tensile test was conducted for neat ABS and the ABS+GNP nanocomposite using a computerized universal tensile testing machine (Make: ZWICK ROELL, Z020, Loadcell 20 KN) at the strain rate of 1 mm min$^{-1}$, with the test speed of 50 mm min$^{-1}$. The stress versus strain values obtained from the sample tensile test were plotted, as shown in Figure 4a.

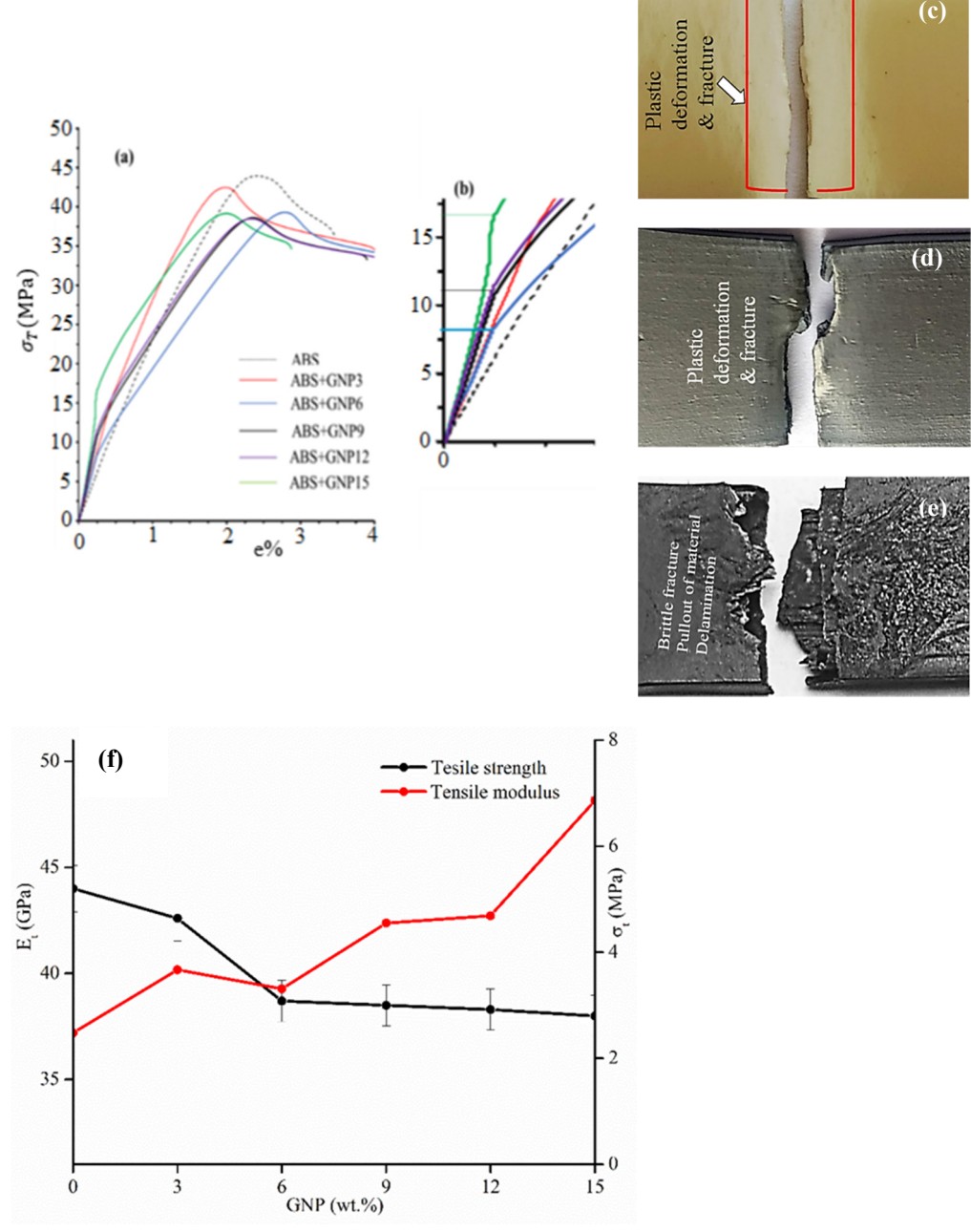

**Figure 4.** Stress–strain plot (**a**,**b**); tensile-fractured-specimen digital photographs of neat ABS, ABS+GNP3, and ABS+GNP15 nanocomposites (**c**–**e**); (**f**) tensile strength and Young's modulus versus GNP concentrations.

The stress–strain plot reveals that the neat ABS and ABS+GNP3 nanocomposite showed a linear relation between stress and strain, up to the peak load. The ABS+GNP6, ABS+GNP9, ABS+GNP12, and ABS+GNP15 nanocomposites showed different behavior and found an initial yielding before the peak load, and further continued loading on these nanocomposites led to a high deformation rate (as seen in magnified Figure 4b). The ABS nanocomposite containing 6–15 wt.% GNPs showed reduced stiffness after the initial yielding, i.e., deviation of the stress–strain curve with decreased slope (as compared to the initial), and a high plastic-deformation rate after early yielding. The bundles of GNPs stacked one above the other in the ABS matrix started slipping against each other due to tensile loading because of the low coefficient of friction between them and their

self-lubrication property. This mechanism led to yielding of ABS+GNP nanocomposites at a lower load, and further continued loading showed a drastic reduction in stiffness [24,25].

The digital photographic images of the tensile-fractured surfaces of ABS, ABS+GNP3, and ABS+GNP15 are shown in Figure 4c–e, respectively. They reveal the ductile fractures in the neat ABS and ABS+GNP3 nanocomposites. The brittle fracture was noticed in the ABS+GNP15 nanocomposite sample. For further microscopic investigation, the neat ABS and ABS+GNP nanocomposite fractured samples were investigated using a scanning electron microscope (SEM). It revealed the cohesive matrix failure of neat ABS, as shown in Figure 5a,b. The uniform dispersion of GNPs in the ABS+GNP3 nanocomposite is observed in the fractured surface morphology shown in Figure 5c,d. Figure 5e,f reveals stacked GNPs and the pull-out of GNPs from the matrix of the ABS+GNP15 nanocomposite. This is due to poor matrix wetting and agglomeration of platelets at higher concentrations. This leads to poor load transfer between filler and matrix. The self-lubricating, loosely packed GNPs lead to slipping, which deteriorates the tensile strength and stiffness after initial yielding.

Figure 4f shows the tensile strength and Young's modulus of ABS+GNP nanocomposites at various concentrations. It was found that the addition of GNPs enhanced both the elastic modulus and strain rate of the ABS. The ABS with 15 wt.% GNP nanocomposite showed the highest modulus of 6.86 GPa, which is increased from 2.48 GPa in neat ABS. The remarkable increase in stiffness of ABS+GNP nanocomposites was due to the reinforcement of high-stiffness GNP fillers, which contributes its stiffness to the resulting nanocomposites [26].

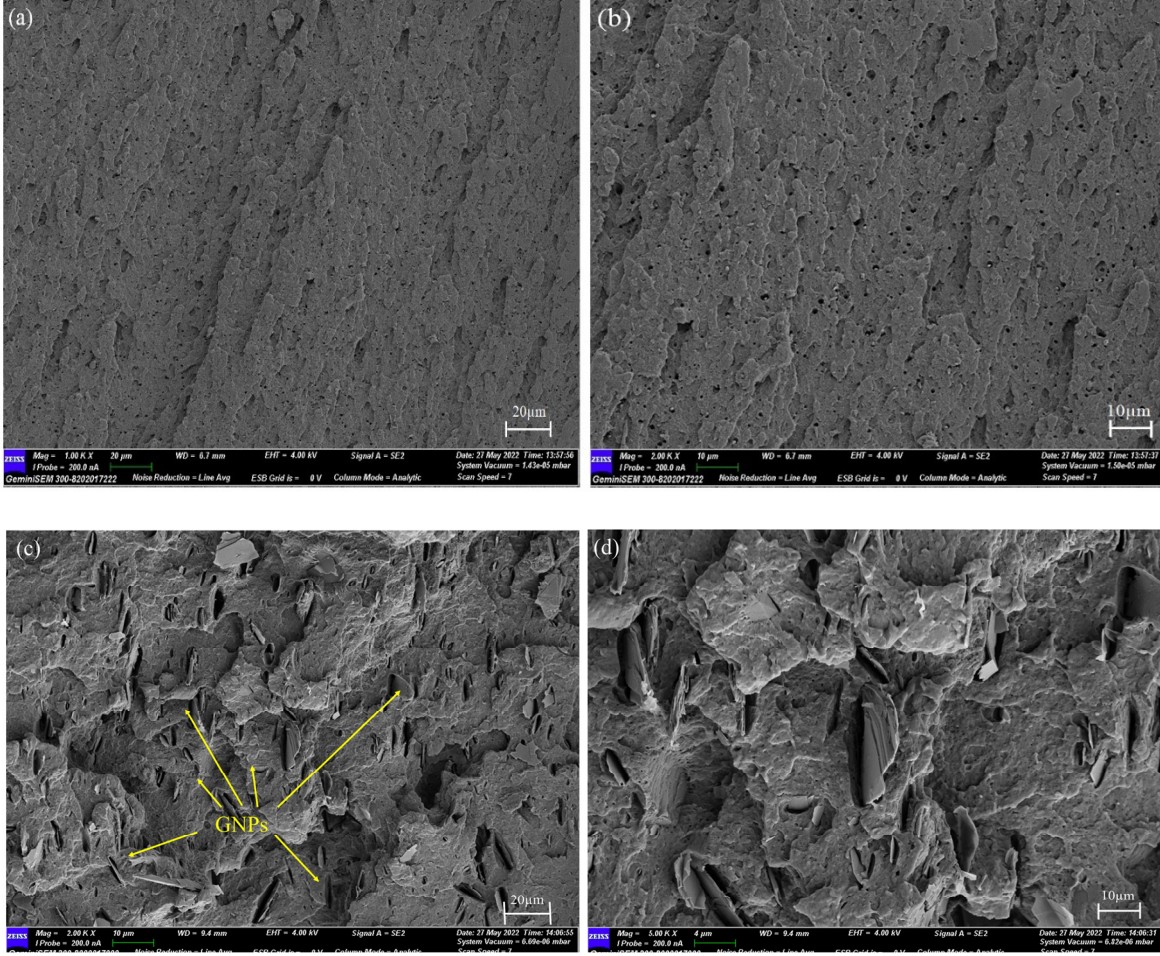

**Figure 5.** *Cont.*

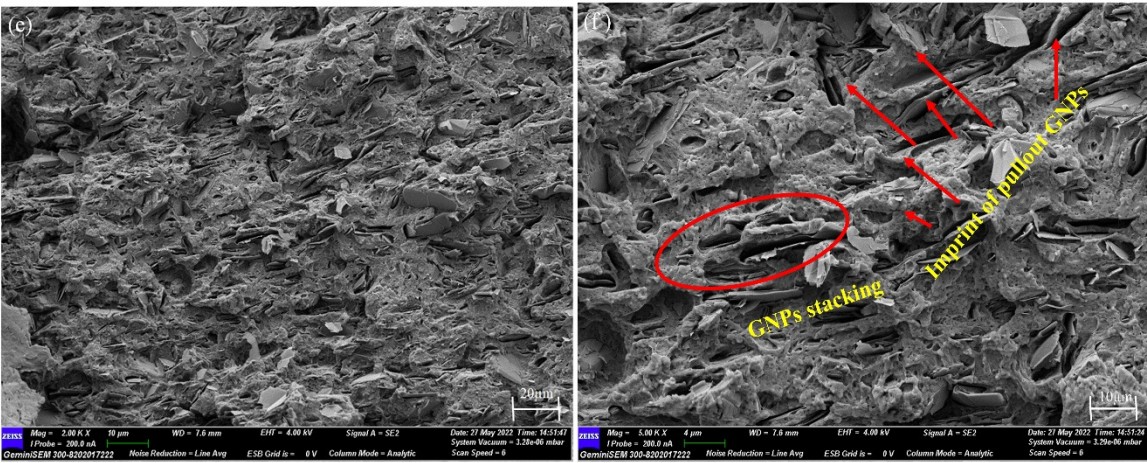

**Figure 5.** SEM images of tensile-fractured-specimen surface of (**a**,**b**) neat ABS; (**c**,**d**) ABS+GNP3; and (**e**,**f**) ABS+GNP15 nanocomposites.

### 3.3.2. Flexural Properties of Composites

The injection-molded pure ABS and AB+GNP nanocomposites were examined under a three-point bending test, and the flexural behavior of the composites is depicted in Figure 6a as a flexural stress versus strain plot. The influence of GNP concentration on flexural strength and modulus of the nanocomposites is depicted as shown in Figure 6b.

The flexural stress was directly proportional to strain up to the peak load (Figure 6a). It is found that the flexural strength was almost constant, and no change was observed due to the addition of GNPs in ABS (Figure 6b). The GNPs have strong in-plane structural strength. However, they have a low coefficient of friction, limited lateral dimension, and a van der Waals force of attraction between the individual layers, restricting the contribution of the in-plane strength of the GNPs to the resultant nanocomposites. The multilayer GNPs, in the processing of nanocomposites, restrict the matrix flow between the layers and lead to poor bonding between the matrix and filler. Hence, the strength of the matrix remains the same. However, due to the strong in-plane strength of fillers and their high stiffness, the flexural modulus of nanocomposites increased with increased GNP content. The flexural modulus of pure ABS is 1.84 GPa, which increases to 3.45 GPa with the addition of 15 wt.% GNPs in the composites.

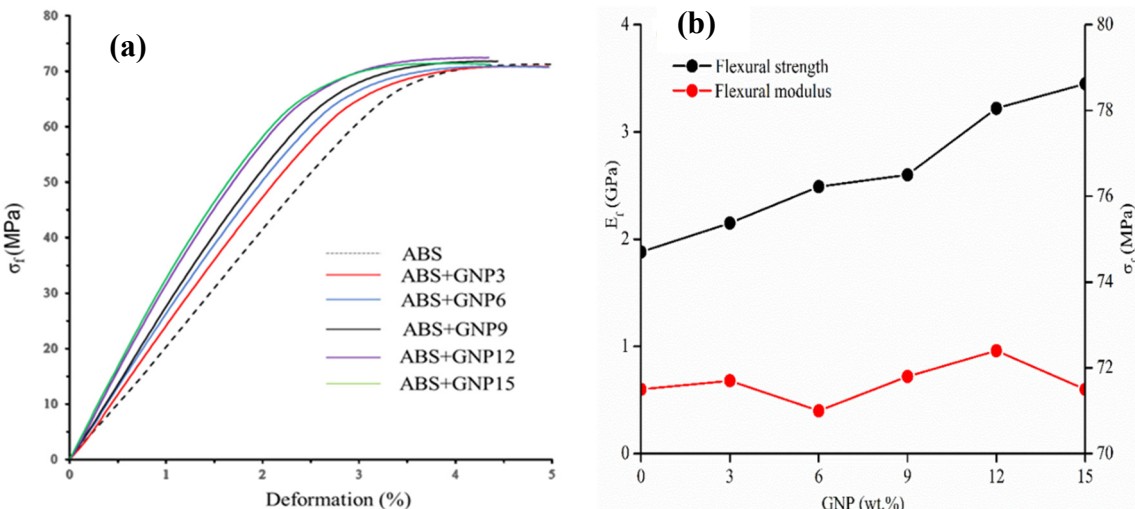

**Figure 6.** (**a**) The plot of flexural behavior of neat ABS and ABS+GNP composites. (**b**) Flexural strength and modulus of ABS and ABS+GNP nanocomposites.

### 3.3.3. Impact Strength

The Izod impact test for neat ABS and ABS+GNP nanocomposites was performed using a Zwick Roell HIT 50P pendulum impact tester. The impact velocity of 3.5 m s$^{-1}$ was maintained for all the samples. Five identical samples were tested in each type of nanocomposite, and average readings were considered and plotted, as shown in Figure 7.

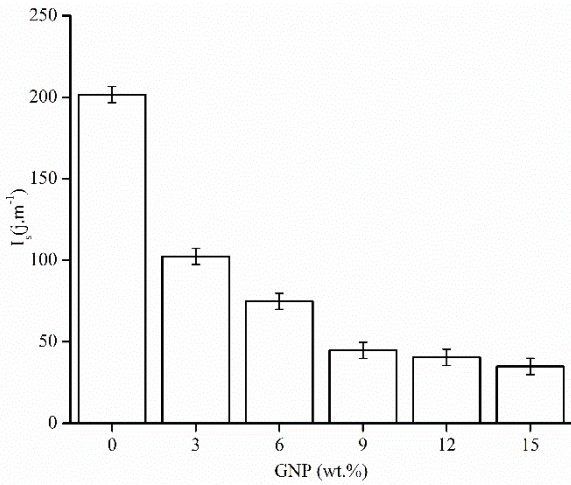

**Figure 7.** Izod impact strength versus GNP concentration in the composites.

It is found that the impact strength remarkably decreased with increased GNPs in the nanocomposites. This indicates that the energy absorption decreased as the filler loading increased. Due to the addition of GNP fillers to the ABS matrix, the resultant nanocomposites became brittle. The incorporation of GNPs into a tough ABS matrix alters the property of the resultant material from ductile to brittle. The energy-absorbing mechanism, in the case of pure ABS, is due to elastic deformation, whereas the addition of GNPs alters the energy-absorption mechanism from ductile deformation to brittle failure, i.e., crack propagation and fracture. Due to the impact load, the rigid and stiff GNPs act as crack imitators in the ABS matrices. These results are in line with the results obtained by various researchers for PC+ABS+GNP nanocomposites [27], HDPE+GNP nanocomposites [28], and PP+graphite composites [29]. The tensile and impact test results found that adding 3 wt.% GNPs into ABS remarkably deteriorates impact resistance but produces moderate tensile and flexural properties compared to neat ABS.

### 3.3.4. Thermal Stability

Thermal stability is an essential property required to be considered from the point of view of processing and application of nanocomposites. Figure 8a depicts the thermogravimetric analysis (TGA)curves of the ABS and ABS+GNP nanocomposites tested under a nitrogen atmosphere. The decomposition of ABS+GNP nanocomposites is a two-step process; the first step is in the range of 380–400 °C (as seen in Figure 8b), and the second step is in the range of 400–475 °C, as shown in Figure 8c. The second step of decomposition perhaps happened due to the interaction between GNPs in ABS polymer. The corresponding temperatures at which 20, 50, and 80% of sample weight loss occur are tabulated in Table 2. It is found that the increased content of GNPs in the ABS improved thermal stability. The TGA curves shift to higher temperatures with increasing GNPs loading, as shown in Figure 8b. The addition of 15 wt.% GNPs led to a slight 3 °C increase in the temperature at which 20% of weight loss is observed. The increase of the temperatures at which there is 50% of weight loss was found to be 5 °C, and a further 9 °C corresponding to 80% weight loss was noticed. The percentage of residual at 600 °C increases as the GNP concentration increases in the ABS matrix, as tabulated in Table 2. However, a marginal improvement in decomposition temperatures at 20 and 50% weight loss is seen. Hence, we can conclude

that marginal improvement of thermal stability was noticed due to the addition of 12 wt.% GNPs and a further 15 wt.% improved thermal stability significantly.

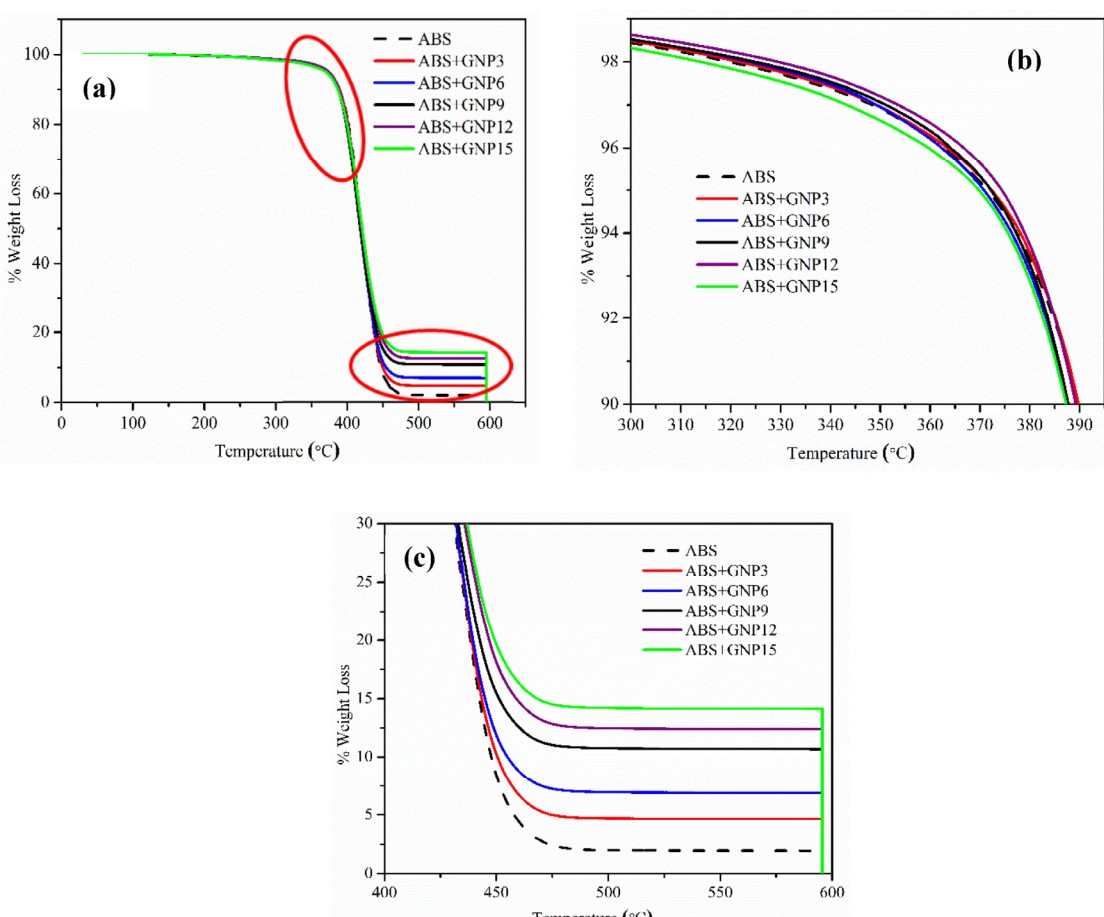

**Figure 8.** TGA curves of ABS and ABS+GNP nanocomposites.

**Table 2.** Thermal degradation characteristics of neat ABS and ABS+GNP composite films.

| Samples | GNPs (wt.%) | 20 wt.% Loss at Temperature (°C) | 50 wt.% Loss at Temperature (°C) | 80 wt.% Loss at Temperature (°C) | Improvement in wt.% Loss Temperature (°C) | | | Residual (%) at 600 °C |
|---|---|---|---|---|---|---|---|---|
| | | | | | at 20 wt.% | at 50 wt.% | at 80 wt.% | |
| ABS | 0 | 399.45 | 420.26 | 438.60 | | | | 2.16 |
| | 3 | 399.45 | 419.84 | 439.35 | 0 | 0.42 | 0.75 | 4.55 |
| | 6 | 396.45 | 419.05 | 439.89 | 3 | 1.21 | 1.25 | 6.58 |
| ABS+GNP | 9 | 400.85 | 420.45 | 444.09 | 1.4 | 0.19 | 5.49 | 10.71 |
| | 12 | 402.25 | 421.85 | 442.69 | 2.8 | 1.59 | 4.09 | 11.30 |
| | 15 | 402.42 | 424.65 | 448.28 | 2.97 | 4.39 | 9.68 | 14.22 |

### 3.4. Electrical Conductivity

The electrical conductivity (σ) of the ABS+GNP nanocomposite films was measured by the two-probe method. The σ as a function of filler concentration was plotted, as shown in Figure 9. The neat ABS showed poor σ, which slightly increased with the addition of GNPs up to 9 wt.%. It is observed that the 12 wt.% GNP-filler-reinforced ABS showed a significant increase in σ compared to 9 wt.%. It is found that the 12 wt.% GNP concentration is the critical filler concentration ($\emptyset_c$) at which the percolation network has been formed in the GNP-reinforced ABS. It is noticed that 9 wt.% to 12 wt.% is the percolation threshold at which the ABS nanocomposite transitions from insulating towards conductive.

Further, increased filler concentration beyond 12 wt.% leads to agglomeration. This produces a slight increase in σ due to saturation. The dispersion of 9–12 wt.% GNPs in the ABS matrix significantly increases the electrical conductivity (σ). The dispersed GNPs in the ABS matrix at critical filler concentration forms micro- and nano-capacitor structure, as shown in Figure 10b. The energy required by the electrons to overcome the insulating thin ABS layer in between GNPs is proportional to the surface area of GNPs and the distance between them. At critical concentration, the distance between dispersed GNPs in the nanocomposite is smaller compared to nanocomposites with low filler concentration ($<\varnothing_c$). In addition, the platelet structure of GNPs having high surface area and aspect ratio promotes the conducting electrons tunneling through the ABS, which increases the conductive path in the nanocomposite. Moreover, GNPs are oriented structures in the vertical and horizontal planes, referred to as Vertical GNP and Horizontal GNP (as seen in Figure 10d), and are responsible for the further increase in conductivity in the 15 wt.% filler-reinforced ABS composites.

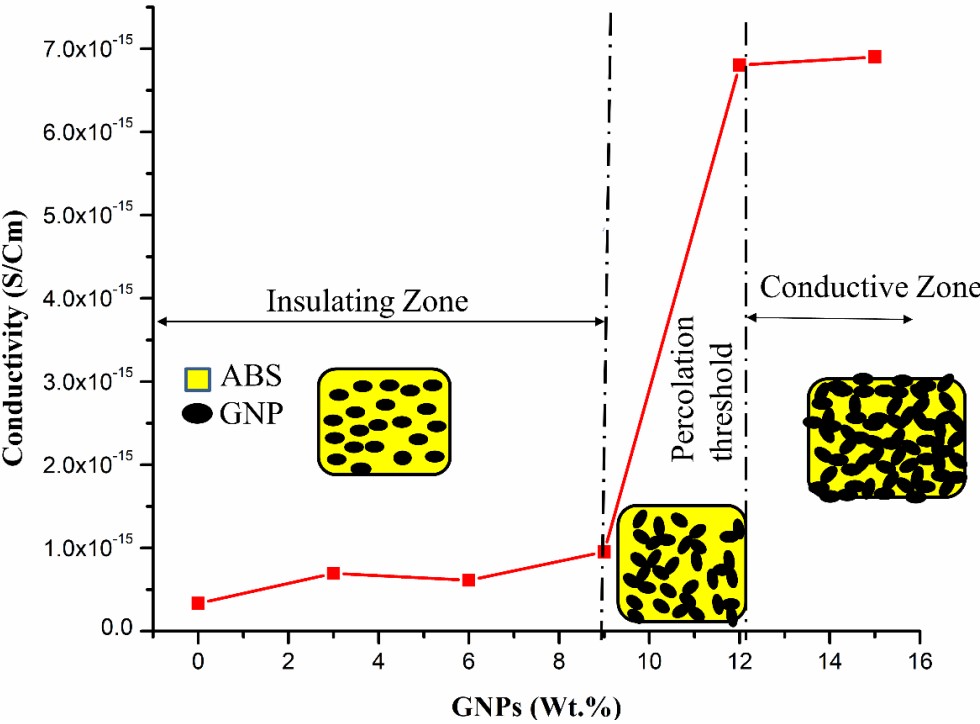

**Figure 9.** Electrical conductivity of the ABS+GNP nanocomposites.

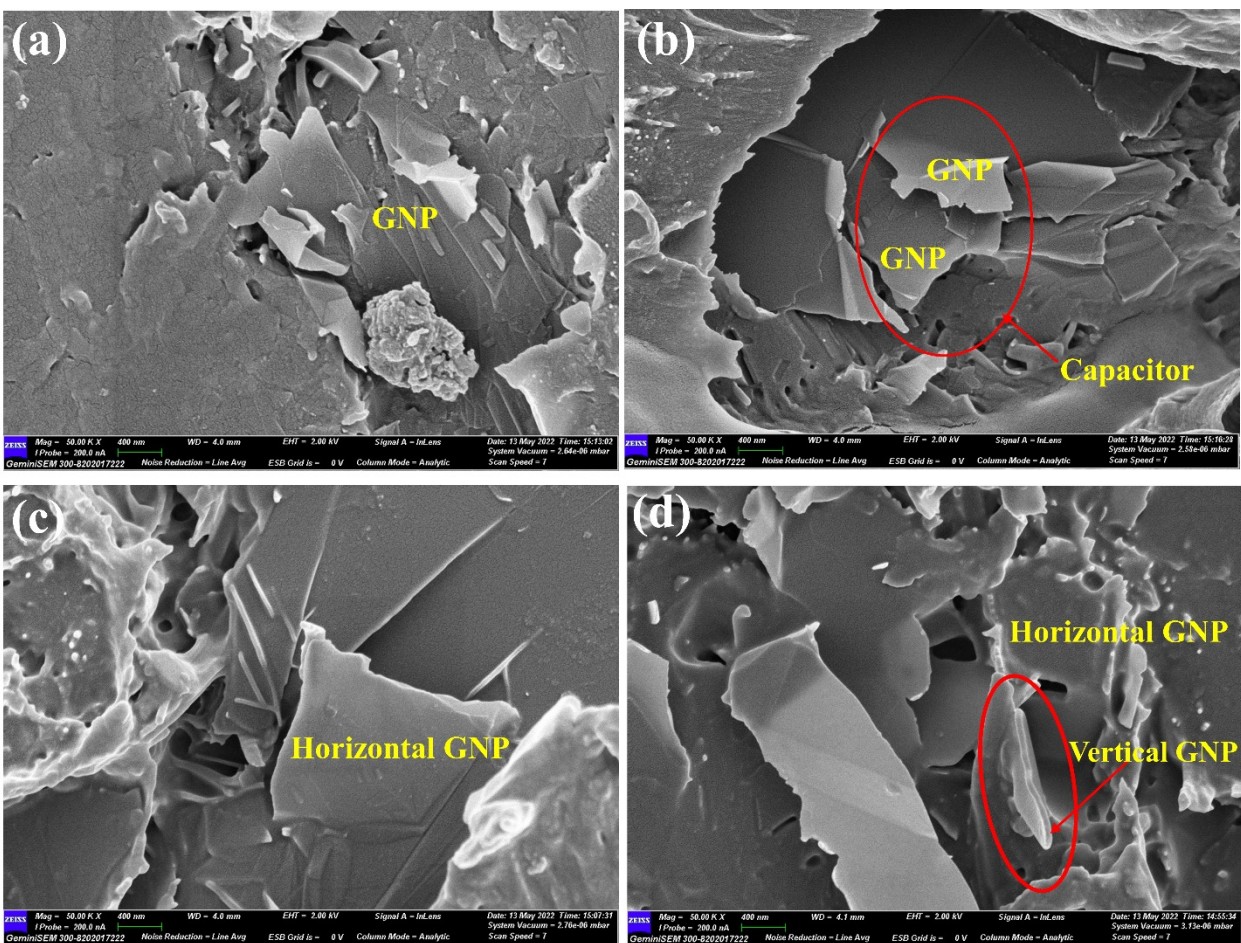

**Figure 10.** The SEM images of polymer nanocomposite films (**a**) ABS+GNP6, (**b**) ABS+GNP9, (**c**) ABS+GNP12, (**d**) ABS+GNP15.

### 3.5. Complex Permittivity, Complex Permeability, and Dielectric and Magnetic Loss of ABS+GNP Nanocomposite Films

The real and imaginary parts of complex permittivity ($\varepsilon'$ and $\varepsilon''$), respectively; complex permeability ($\mu'$ and $\mu''$), respectively; dielectric loss (tan $\delta_\varepsilon$); and magnetic loss (tan $\delta_\mu$) are essential properties for the electromagnetic-wave-shielding characteristics of nanofiller-reinforced polymer composites, in addition to electrical conductivity ($\sigma$). The $\varepsilon'$ is the ability of the material to store the energy, and $\varepsilon''$ is the electrical-energy loss capability of the material. Similarly, the $\mu'$ and $\mu''$ are the amount of energy stored and energy loss in the material due to the magnetic field, respectively. These properties are the functions of conductivity; filler concentration; and operating frequency. In the present work, the complex permittivity and permeability of ABS+GNP nanocomposite films were measured by the waveguide method using a vector network analyzer. The $\varepsilon'$, $\varepsilon''$, $\mu'$, $\mu''$, tan $\delta_\varepsilon$, and tan $\delta_\mu$ of all the ABS+GNP nanocomposites as a function of frequency at the X-band are shown in Figure 11a–f.

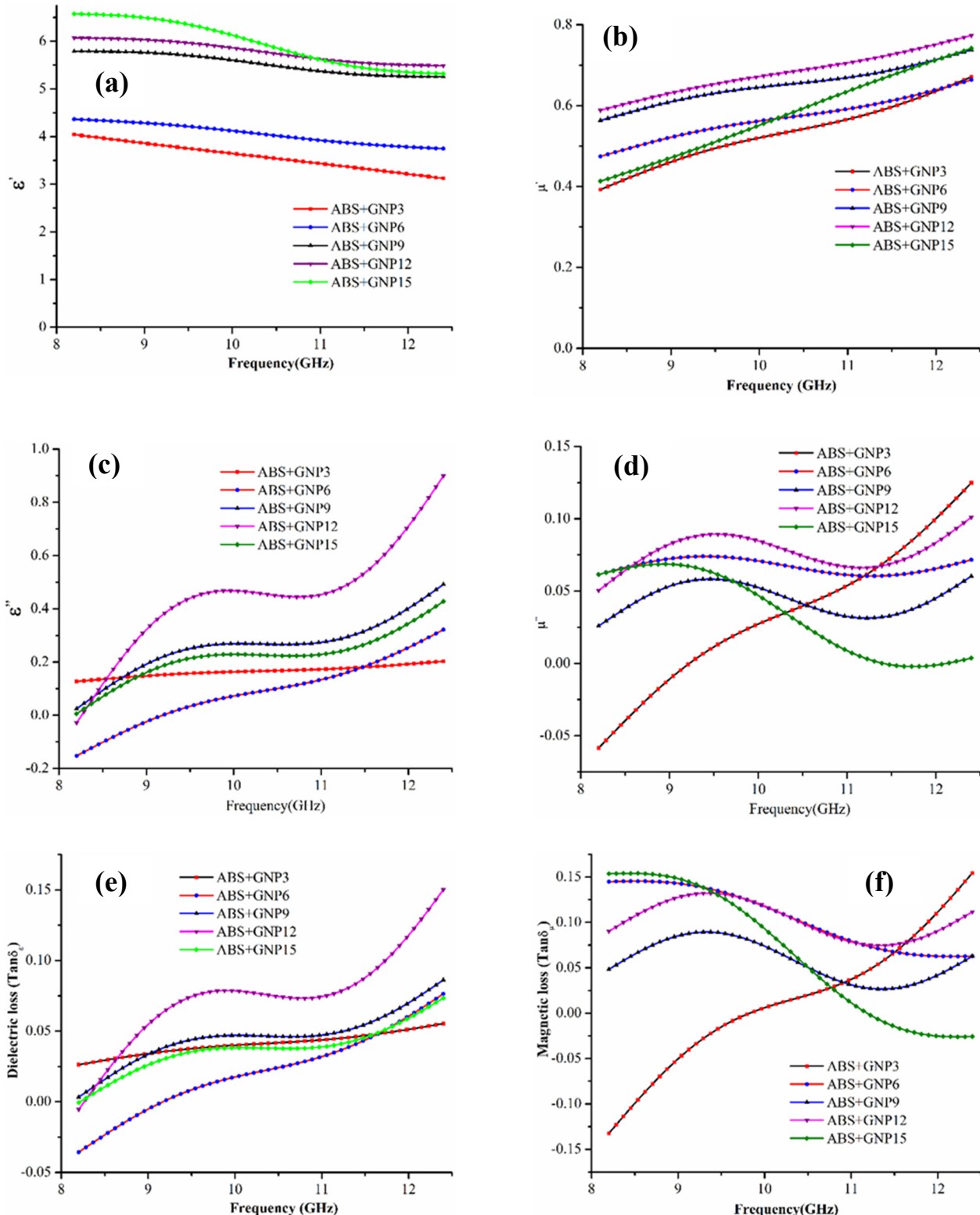

**Figure 11.** Frequency-dependent behavior of the ABS+GNP nanocomposites for (**a**) real and (**c**) imaginary parts of permittivity, (**b**) real and (**d**) imaginary parts of permeability, and corresponding (**e**) dielectric and (**f**) magnetic losses.

It was found that the addition of GNPs to the ABS matrix improved the $\varepsilon'$. This is due to the formation of the microcapacitor and the presence of layered-structure GNPs in the ABS matrix. The GNPs act as nanoelectrodes in between GNPs and ABS, which acts as

a nanodielectric. This type of structure increases with an increase in GNP concentration in the ABS matrix. The distance between dispersed GNP surfaces in the ABS matrix reduces as the concentration of GNPs increases. This is depicted in the microstructure study of ABS+GNP films, as shown in SEM images in Figure 10d. A higher number of nanostructured capacitors with a thin layer of ABS matrix was observed at higher concentrations, as shown in Figure 5d. As a result, a remarkable improvement of $\varepsilon'$ at higher concentrations of the GNPs was observed. Also, the increase in the $\varepsilon'$ agrees with the conductivity of the nanocomposite. It is also noticed the $\varepsilon'$ decreases as the operating frequency increases [30], as shown in Figure 11a for all the ABS+GNP nanocomposites. This decrease is due to the reduction of the space charge polarization effect.

The $\varepsilon''$ represents the electrical-energy loss capability of the material. Low and high values of imaginary permittivity are preferred for charge storage and EMI-shielding applications, respectively [31]. In the present work, the addition of 3 wt.% GNPs in the ABS matrix showed $\varepsilon''$ is independent of frequency, whereas the 9–12 wt.% GNP-filled ABS showed higher $\varepsilon''$ and depended on frequency in X-band range. The $\varepsilon''$ varied from 0.12–0.2 for ABS+GNP3, but other samples varied from 0.2–0.9 at 12.4 GHz, as shown in Figure 11c. The 15 wt.% GNPs in the ABS had $\varepsilon''$ decrease to 0.4. The increase in $\varepsilon''$ is observed in the nanocomposite with increased content of GNPs, as shown in Figure 11c. This is due to increased conductivity of the nanocomposite, which is achieved by the three-dimensional network structure of GNP dispersion in the ABS matrix (as seen in Figure 10d). Moreover, the addtion of 15 wt.% GNPs in the ABS showed an increase in the $\varepsilon''$ due to the dipole moment and conductivity of ABS nanocomposites [32]. The relation between the imaginary part of permittivity ($\varepsilon''$) and conductivity ($\sigma$) is known as the Debye relation and can be written as [33,34]

$$\varepsilon'' = \frac{\sigma}{2\pi f \varepsilon_0} + \varepsilon''_{relax} \qquad (2)$$

where $\sigma$ is the electrical conductivity, $\varepsilon_0$ is the free-space permittivity, and $\varepsilon''_{relax}$ signifies the relaxation loss. The above equation indicates the importance of conductivity in the dielectric behavior of the EMI-shielding material. The $\varepsilon''$ ascribed to conductance loss $\left(\frac{\sigma}{2\pi f \varepsilon_0}\right)$ and relaxation loss, indicates higher conductivity is favorable to improving the $\varepsilon''$.

The $\mu'$ and $\mu''$ relate to magnetic-energy storage and loss. The increase in the real part of the permeability ($\mu'$) tends to increase with the filler concentration and the frequency due to an increase in magnetic loss, as shown in Figure 11b. The $\mu'$ increases from 0.47–0.6 at 8.2 GHz and 0.66–0.77 at 12.4 GHz. The addition of 15 wt.% GNPs in the ABS increases the magnetic loss linearly with the frequency. The magnetic loss increases from −0.2–0.15 at 8.2 GHz and 0.13–0.08 at 9.2 GHz and decreases from 0.02 to −0.02 between 9.2–11.2 GHz. Further, it increases with the increase in frequency, but at the addition of 3 wt.% GNPs, the magnetic loss varies from −0.2 to 0.15 and increases with the frequency, as shown in Figure 11f.

The imaginary part of the permeability of the ABS+GNP3 nanocomposite film has an increasing trend with frequency. It can be attributed to the coupling of non-magnetic material with the magnetic component of the EM wave at high frequency. The negative value of $\mu''$ for the ABS+GNP3 composite in the low-frequency region (8.2–9.2 GHz) may be due to polariton resonance excited in the material [33]. Furthermore, increased GNPs in the ABS nanocomposites were independent of frequency due to the non-uniform dispersion of GNPs.

## 4. Electromagnetic-Shielding Effectiveness

Electromagnetic waves interact with the shielding material and get attenuated due to absorption (SE$_A$), reflection (SE$_R$), and multiple reflections (SE$_{MR}$) [35]. The SE$_R$ occurs on the surface of the shielding material due to impedance mismatch [36]. The SE$_A$ is contributed by the internal geometrical structure and functional properties of nanoparticles in the nanocomposite. SE$_A$ leads to the generation of heat due to the conversion of EM wave energy into heat. The signals produce successive reflections from the different interfaces

within the matrix, leading to SE$_{MR}$. However, the SE$_{MR}$ is small compared to the other two types of attenuation; therefore, it is negligible.

The electromagnetic-interference-shielding effectiveness (EMI SE) of the ABS+GNP nanocomposites was investigated using a two-port N9911A vector network analyzer (Keysight Technologies, Bangalore, India). The measurement setup consisted of a vector network analyzer connected to an X-band waveguide (WR-90) through an adopter using a coaxial cable, as shown in Figure 12. The ABS and ABS+GNP nanocomposite samples were accurately prepared to 22.86 mm × 10.16 mm and placed in the sample holder between the two waveguides. The reflection and transmission (S11 and S12) scattering parameters were measured for X-band-range frequencies. Before the measurement, the vector network analyzer calibration was performed. The SE$_A$, SE$_R$, and total electromagnetic-interference-shielding effectiveness (SE$_T$) can be evaluated from the S11/S22 and S12/S21 parameters using Equations (6)–(8) mentioned below and plotted against frequencies in the X-band.

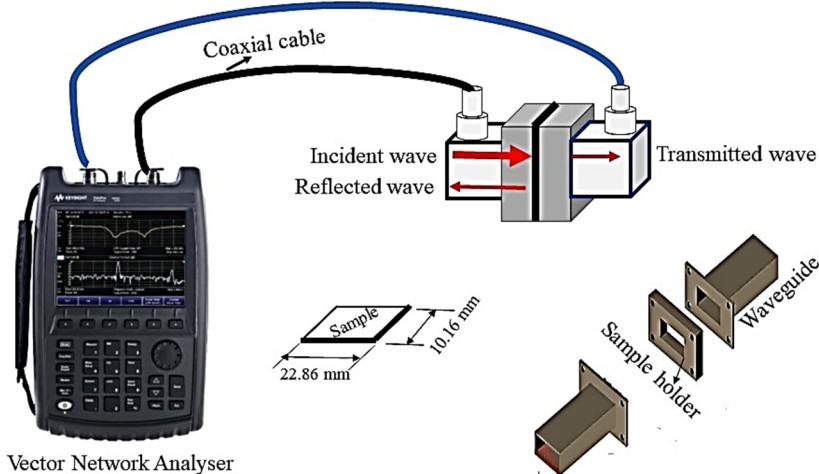

**Figure 12.** Network-analyzer experimental setup to measure S-parameter.

$$SE_T = SE_A + SE_R + SE_M \tag{3}$$

$SE_T = SE_A + SE_R$ If $SE_A > 10$ dB, $SE_M$ is neglected

$$R = |S_{11}|^2 = |S_{22}|^2 \tag{4}$$

$$T = |S_{12}|^2 = |S_{21}|^2 \tag{5}$$

$$SE_T(db) = 10log_{10}\left(\frac{1}{S_{12}^2}\right) = 10log_{10}\left(\frac{1}{S_{21}^2}\right) = log_{10}\left(\frac{1}{T}\right) \tag{6}$$

$$SE_R(db) = 10log_{10}\left(\frac{1}{1-S_{11}^2}\right) = 10log_{10}\left(\frac{1}{1-R}\right) \tag{7}$$

$$SE_A(db) = 10log_{10}\left(\frac{1-S_{11}^2}{S_{12}^2}\right) = 10log_{10}\left(\frac{1-R}{T}\right) \tag{8}$$

EM waves passing through the neat ABS showed negligible shielding effectiveness due to almost insignificant reflection and absorption, as shown in Figure 13a–c. This is due to no charge carriers in the ABS matrix, which is highly transparent to EM waves. The shielding effectiveness increases with the increase of GNP content in the nanocomposites. With the increased GNP concentration in the ABS matrix, more GNPs interact with the EM waves,

which increases the attenuation of EM waves in both reflection and absorption. The highest shielding effectiveness was noticed at 10–10.5 GHz frequency. The order of the EMI SE$_T$ of nanocomposites is ABS+GNP15 > ABS+GNP12 > ABS+GNP9 > ABS+GNP6 > ABS+GNP3. The highest shielding effectiveness was achieved at 30.74 dB for ABS+GNP15, 28.66 dB for ABS+GNP12, 25.59 dB for ABS+GNP9, 10.97 dB for ABS+GNP6, and 5.79 dB for ABS+GNP3 composites at 10–10.5 GHz. The 6 to 15 wt.% GNP-loaded nanocomposites showed more EM-wave-absorption dependency on frequency than that for reflection dependency. The dependency of the EMI SE$_T$ and EMI SE$_R$ on frequency is less in the case of the ABS+GNP3 nanocomposite.

The ABS+GNP nanocomposite surface has charge carriers that interact with EM waves and primarily reflect the EM waves. The reflected EM waves are not completely weak but have some radiation effect; hence, shielding by reflection generates secondary EMI pollution. An effective EMI-shielding material has high absorption of EM waves rather than reflection. Hence, the effect of GNP concentration on the reflection and absorption of nanocomposites in dB and percentage is shown in Figure 13d,e, respectively.

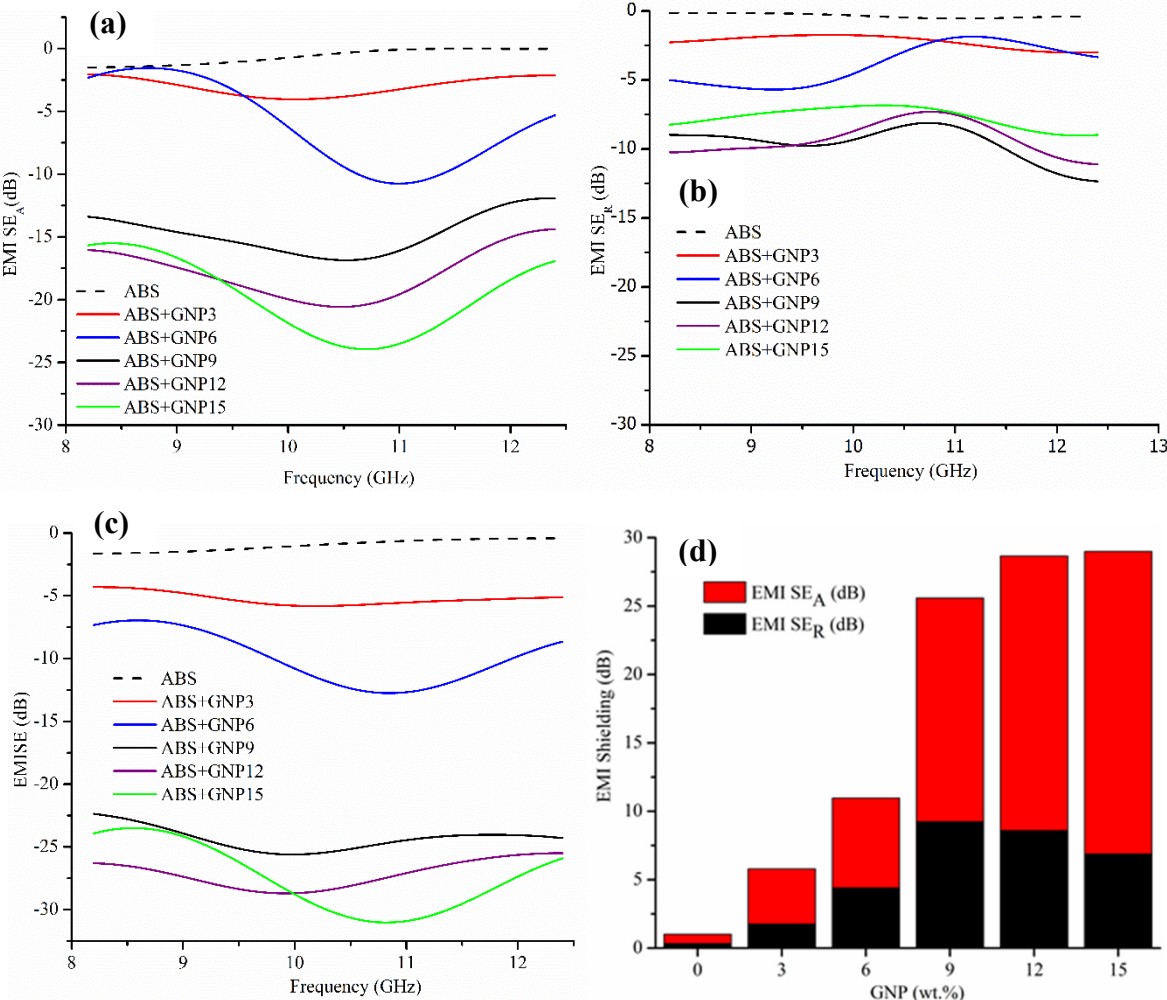

**Figure 13.** *Cont*.

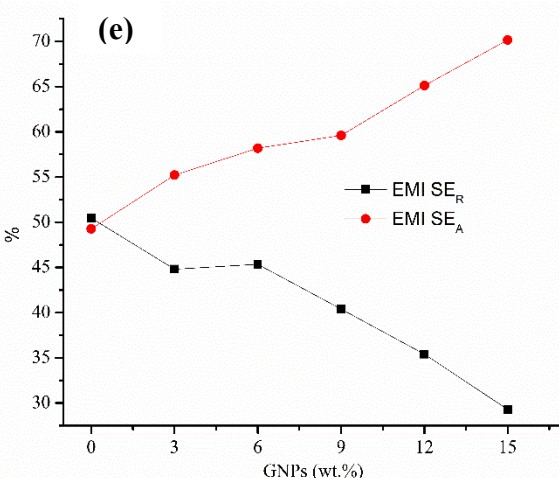

**Figure 13.** EMI SE plots in X-band: (**a**) $SE_A$, (**b**) $SE_R$, and (**c**) $SE_T$. (**d**) Maximum absorption and reflection at 10 GHz. (**e**) Percent reflection and absorption of ABS+GNP nanocomposites.

It is observed in Figure 13d that reflection and absorption were increased with the concentration of GNPs, up to 9 wt.%. The increase is due to the higher availability of mobile charge carriers at the surface of concentrated GNPs. Further increase of GNPs in the composites enhanced the absorption continuously and decreased reflection. This is attributed to the three-dimensional network structure formed in the ABS matrix at a higher concentration of GNPs [37]. In such nanocomposites, the EM waves enter the composite through a three-dimensional network, and their energy is converted into heat. Further, the GNPs also have good thermal conductivity, leading to heat dissipation, and the rate of increase in absorption was higher than reflection with respective GNP concentration [38]. The attenuation of EM radiation incident on nanocomposites containing 9–15 wt.% GNPs showed a lower rate of reflection, as shown in Figure 13e. The ABS+GNP15 nanocomposite showed the lowest reflection and the highest absorption, so it had the best shielding effectiveness compared to other samples.

## 5. Conclusions

ABS nanocomposites were prepared with 3 to 15 wt.% GNPs in steps of 3 wt.% by compounding, extrusion, and the compression-molding method. Their mechanical and thermal properties and electromagnetic –shielding effectiveness were investigated, and this study's conclusion is obtained by the results listed below.

- The modulus of the ABS improved due to the addition of high-stiffness GNPs. The slightly decreased tensile strength was noticed due to interruptions of the intermolecular structure of ABS due to GNPs in the composite. Further, it deteriorates at a higher concentration of GNPs due to poor wetting and agglomeration. In addition, the impact strength is reduced, and the composite becomes brittle due to the addition of GNPs.
- The thermal stability of ABS improved through the thermal barrier, and a faster heat-conducting path developed due to the addition of GNPs.
- The better dispersion state is achieved between 9–12 wt.% GNPs, at which nanocomposites transit from insulate to conductive.
- The addition of GNPs to the ABS matrix improves the $\varepsilon'$, which decreases with the increase in operating frequency. The $\varepsilon''$ is independent of frequency for ABS+GNP3, and 9–12 wt.% GNP-filled ABS is dependent on frequency.
- The dielectric loss increases as filler concentration increases, whereas magnetic loss increases at a lower frequency and decreases at higher frequencies.
- The electromagnetic shielding of ABS improved due to the addition of GNPs, and absorption is the dominating mechanism in the composites consisting of 9–15 wt.% GNPs. It is found that 15 wt.% GNP-filled ABS composites show the highest EMI-

shielding effectiveness compared to other nanocomposites, in addition to moderate mechanical and thermal properties.

- The 1–2 µm GNP (lateral dimension)-filler-reinforced ABS composites may achieve 155 dB/mm shielding effectiveness.

**Author Contributions:** R.B.J.C.: investigation, methodology and original draft preparation. B.S.: Conceptualization and supervision. M.S.K.: formal analysis. N.N.P.: Software. D.S.: Resources for experimentation and data curation. All authors have read and agreed to the published version of the manuscript.

**Funding:** This research received no external funding.

**Data Availability Statement:** Data is included in the manuscript.

**Conflicts of Interest:** The authors declare no conflict of interest.

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
