# Peer review of "Mechanical and Electrical Properties and Electromagnetic-Wave-Shielding Effectiveness of Graphene-Nanoplatelet-Reinforced Acrylonitrile Butadiene Styrene Nanocomposites"

_jcs, doi:10.3390/jcs7030117_

Round 1

Reviewer 1 Report

I would like to recommend the manuscript for publication after a revision, incorporating the following comments:

1. In page 6, the authors mentioned “This behavior of yielding at a lower load and drastic reduction of stiffness after early yielding is due to the slippage of stacked GNPs in the matrix.” Please explain the mechanism further.

2. In page 9, the authors mentioned “The high stiffness of the GNPs incorporation into a tough ABS matrix alters the energy-absorbing mechanism.” Please explain the energy absorption mechanism of pure ABS and ABS+GNPs.

3. Why is the electrical conductivity (σ) of ABS+GNPs15 only slightly increased compared to ABS+GNPs12?

4. In Fig. 11 (a, c and e), the variation rule of electromagnetic parameters is difficult to understand. Please refer to Journal of Materials Science & Technology, 2022, 127, 48-60.

5. The font format of the picture is not uniform, some pictures are elongated, and the picture labels are not aligned. In addition, for example, in Figure 4(a and b), there are two "(a)" labels and two "(b)" labels, respectively. In Figure 4(c and d), the "(c)" and "(d)" labels are not fully displayed. In Figure 4(e), the vertical coordinate is not fully displayed

6. The sorting of subheadings is problematic.

7. Some important references should be highlighted such as Chemical Engineering Journal, 2023, 452, 139042; Carbon, 2021, 177, 412-426.

Author Response

Dear sir

As per your review comments, the changes are incorporated and answered to the review comments and uploaded

Reviewer 2 Report

Mechanical, Electrical properties and Electromagnetic wave shielding effectiveness of Graphene Nano Platelets reinforced Acrylonitrile Butadiene Styrene nanocomposites

Journal of Composite Science

The paper is well written and structured.

I found some details which need to be fixed (see paper enclosed).

The quality of the figures should be improved.

The paper reports data obtained from a Vector Network Analyzer. This technique allows to measure the EMI shielding capacity at high frequency (10GHz). To put in perspective the work realized, it would be useful to compare the shield efficiency of 25% (Figure 13) with the unique one reported (60 dB) in page 2. Are the 60 db a sum of SEa and SEr or is it SET? What is the current art of shield efficiency? (At least the most used material and shield efficiency in dB)?

Here are several suggestions:

·       Page 2. This paragraph should be erased because of the lack of quantitative data and because the raw materials carried out are graphite flakes not graphene:

Graphene is one of the 2D structured nanomaterials with attractive properties such 56 as good mechanical strength, excellent carrier mobility, high thermal conductivity, and 57 thermal stability. It also has a high melting point of 2000 °C[13]. Because of the above said 58 attractive properties, Graphene has been used as a filler for several polymers for various 59 applications.

·       The literature survey lacks of data on EMI shielding data. I would suggest to explain what are the targets in both electrical conductivity and mechanical properties (quantitative values).

·       Page 3. Figure 1. I did not understand the picture

·       Page 4. Could you specify what S-parameter stands for?

·       Page 4. X-ray diffraction reveals the presence of graphite not graphene.

·       Page 5. The details of beta, the wavelength, K should be given.

·       A shift of the peaks to the low angle is observed for ABS+GNP9. How do you explain such shift?

·       Page 6. The fracture observations are very convincing. Would you have some idea of the reproducibility of the tensile testing between 2 different specimens?  

·       Page 7. Add a scale of the images.

·       Page 9.  TGA curves: what is the chemical composition of the residue obtained at 600°C under N2 ?

·       Page 10. Figure 10: scale of the image and improve the caption by reporting the meaning of V and P…

·       Page 12. Figure 10 : scale and how did you identify GnP from ABS?

If you could add the ABS matrix in the figure 11 it will be useful…

·       Page 14. Merge equation (2) and (3), define all the parameters. I do not understand ε’’relax

·       Page 17. three-dimensional network structure formed in the ABS matrix at a higher concentration of GNPs. The SEM observation carried out in the paper Figure 10 suggest that there are V and P GnP at high concentration : did I understand wright?

Author Response

Dear sir

As per your suggestions, the changes are made, the reviewer questions are answered, and the file is uploaded

Round 2

Reviewer 1 Report

The authors have addressed my comments and I recommend the publication of this revised manuscript.

Author Response

Response to Reviewer 1 Comments

Point 1: English language and style are fine/minor spell check required.

Response 1: The spell check has done .

Reviewer 2 Report

Dear Authors, you did not answer to my question concerning the current art of EMI shield. To my question, you answered:

Response 1: Sachdev et al. [5]  developed microsize (10-20µm) graphite filler-loaded ABS composites. The dry mixing followed by the hot compression molding method and manufactured 3mm thick plates and investigated EMISE and achieved 60dB for 3mm thickness samples.

In this research work, the nanofiller size(1-2 µm) and developed by ABS/GNPs granules through injection molding. Further, nanocomposite films were fabricated by compression molding to produce 200 µm (0.2mm) thickness samples that achieved 30.74dB, whereas 3mm samples achieved 60 dB..

What should I understand ?? With a 3mm sample you achieved 60 dB same as Sachdev et al. [5] ?? A current state of the art of EMI shielding woul have help the reader on the interest of GnP in such field.

Wang et al.[16] inspired by nature and developed a Co3O4@WSe2-MWCNTs nano-micro 78 “vine”with a hierarchical structure and found efficient green EMI shielding. Also, the vine 79 structure performs as a supercapacitor. They constructed a multifunctional microwave 80 conversion and storage device. It converts EM radiation and is stored as useful electrical 81 energy

This paragraph does not help to the question of the current art of EMI shielding.

Best regards

1.66×10−1 S/cm : 0.166 would be better

0.14g-cm-3 : g.cm-3 would be better

Mpa : MPa would be better

Author Response

Response to Reviewer R2 Comments

Point 1: Response 1: Sachdev et al. [5]  developed microsize (10-20µm) graphite filler-loaded ABS composites. The dry mixing followed by the hot compression molding method and manufactured 3mm thick plates and investigated EMISE and achieved 60dB for 3mm thickness samples.

In this research work, the nanofiller size(1-2 µm) and developed by ABS/GNPs granules through injection molding. Further, nanocomposite films were fabricated by compression molding to produce 200 µm (0.2mm) thickness samples that achieved 30.74dB, whereas 3mm samples achieved 60 dB..

What should I understand ?? With a 3mm sample you achieved 60 dB same as Sachdev et al. [5] ?? A current state of the art of EMI shielding woul have help the reader on the interest of GnP in such field.

Response 1:

In the present work, we used 1-2 µm GNP (lateral dimension) fillers reinforced in ABS composites and achieved 30.74 dB shielding effectiveness for 200 µm thickness nanocomposite films. The shielding effectiveness is also dependent on the thickness of the material. Our material is minimal in thickness compared to the available literature of 3mm. Hence, we can conclude samples of ABS/GnP will give better results than the literature value reported.

Point 2: Page 2. Wang et al.[16] inspired by nature and developed a Co3O4@WSe2-MWCNTs nano-micro  “vine” with a hierarchical structure and found efficient green EMI shielding. Also, the vine structure performs as a supercapacitor. They constructed a multifunctional microwave conversion and storage device. It converts EM radiation and is stored as useful electrical energy

Response 2: It is a multifunctional EM shielding material, and its multifunctionality is explained in this paragraph.

Round 3

Reviewer 2 Report

Dear Authors,

please add :

1_in the introduction:  that so far the best or commercial value is xxdB/mm.

2_please add in the conclusion the fact that 1-2 µm GNP (lateral dimension) fillers reinforced ABS composites allow to achieve 155 dB/mm shielding effectiveness.

Great job.

Best regards

R.M.

Author Response

Point 1: 1_in the introduction:  that so far, the best or commercial value is xxdB/mm..

Response 1: If the shielding effectivenes by the material is below 10 dB  it is considered as little or no shielding effectivenes, 10 to 20 dB is considered as minimum effective range. If its morthan 20 dB it will be considered as effective shielding and suitable for commercial application.

Point 2: 2_please add in the conclusion the fact that 1-2 µm GNP (lateral dimension) fillers reinforced ABS composites allow to achieve 155 dB/mm shielding effectiveness.

Response 2: It is added in the conclusion section.

Round 4

Reviewer 2 Report

Good work !